# Clinical, Radiological, and Histopathological Characteristics of Periosteal Chondrosarcoma with a Focus on the Frequency of Medullary Invasion

**DOI:** 10.3390/jcm11072062

**Published:** 2022-04-06

**Authors:** Makoto Nakagawa, Makoto Endo, Yosuke Susuki, Nobuhiko Yokoyama, Akira Maekawa, Akira Nabeshima, Keiichiro Iida, Toshifumi Fujiwara, Nokitaka Setsu, Tomoya Matsunobu, Yoshihiro Matsumoto, Ryohei Yokoyama, Yuichi Yamada, Kenichi Kohashi, Hidetaka Yamamoto, Yoshinao Oda, Yukihide Iwamoto, Yasuharu Nakashima

**Affiliations:** 1Department of Orthopaedic Surgery, Graduate School of Medical Sciences, Kyushu University, 3-1-1, Maidashi, Higashi-ku, Fukuoka 812-8582, Japan; nakagawam59@gmail.com (M.N.); nabeshima.akira.031@m.kyushu-u.ac.jp (A.N.); iida.keiichiro.979@m.kyushu-u.ac.jp (K.I.); fujiwara.toshifumi.771@m.kyushu-u.ac.jp (T.F.); matsumoto.yoshihiro.540@m.kyushu-u.ac.jp (Y.M.); nakashima.yasuharu.453@m.kyushu-u.ac.jp (Y.N.); 2Department of Anatomic Pathology, Pathological Sciences, Graduate School of Medical Sciences, Kyushu University, 3-1-1, Maidashi, Higashi-ku, Fukuoka 812-8582, Japan; susuki.yosuke.585@m.kyushu-u.ac.jp (Y.S.); yamada.yuuichi.957@m.kyushu-u.ac.jp (Y.Y.); kohashi.kenichi.588@m.kyushu-u.ac.jp (K.K.); yamamoto.hidetaka.738@m.kyushu-u.ac.jp (H.Y.); oda.yoshinao.389@m.kyushu-u.ac.jp (Y.O.); 3Department of Orthopaedic Surgery, Kyushu Cancer Center, 3-1-1, Notame, Minami-ku, Fukuoka 811-1395, Japan; hariganenock@gmail.com (N.Y.); sets0rockandsnow@gmail.com (N.S.); ryoheiyoko@nifty.com (R.Y.); 4Department of Orthopaedic Surgery, Kyushu Rosai Hospital, 1-1, Sonekita, Kokuraminami-ku, Kitakyushu 800-0296, Japan; maekawarugby@gmail.com (A.M.); matsunob@ortho.med.kyushu-u.ac.jp (T.M.); iwamoto.yukihide.386@m.kyushu-u.ac.jp (Y.I.)

**Keywords:** periosteal chondrosarcoma, medullary invasion, MRI, wide resection, surgery planning, histologically negative margin

## Abstract

Periosteal chondrosarcoma is an extremely rare malignant cartilage-forming tumour that originates from the periosteum and occurs on the surface of bone. Often, it is difficult to distinguish periosteal chondrosarcoma from other tumours, and reports in the literature are scarce. This study aims to investigate the characteristics of periosteal chondrosarcoma, focusing particularly on medullary invasion. Among 33 periosteal cartilaginous tumours, seven patients with pathologically proven periosteal chondrosarcoma were identified retrospectively. The average tumour size was 5.4 cm in the long axis; two tumours were smaller than 3.0 cm. Six tumours were resected with a wide margin, and the remaining tumour had a marginal margin. Histology revealed that six tumours (85.7%) had invaded the medullary cavity; three of these did not show invasion into the medullary cavity on MRI evaluation. Neither local recurrence nor metastasis was observed among these patients. The frequency of invasion of the medullary cavity was higher than that reported previously. The recommended treatment for periosteal chondrosarcoma is resection with an adequate margin. Therefore, surgeons should consider the possibility of medullary invasion when attempting to achieve a histologically negative margin, even if the tumour does not show invasion into the medullary cavity on MRI.

## 1. Introduction

Periosteal chondrosarcoma is a malignant cartilage-forming tumour that originates from the periosteum and grows on the surface of the bone [1]. The tumour is extremely rare, accounting for less than 0.5% of all chondrosarcomas; it occurs in adults, primarily in the second to fourth decades of life, with a slight male predominance [1]. Typically, it involves the metaphysis of long tubular bones, most commonly the distal femur and the humerus. A diagnosis of periosteal chondrosarcoma is often difficult, although diagnostic criteria are published in the WHO classification of tumours [1]. Differential diagnoses include periosteal chondroma, periosteal osteosarcoma and secondary peripheral chondrosarcoma. Periosteal chondrosarcoma is defined as a malignant tumour, and some studies report local recurrence and pulmonary metastasis after inappropriate excision [2,3,4]. Therefore, accurate diagnosis of periosteal chondrosarcoma and performance of complete resection are needed for a better prognosis.

Periosteal chondrosarcoma arises from the periosteum and, usually, grows on the outside of the bone; however, it can occasionally extend into the medullary cavity [5,6,7,8]. Although the frequency of medullary invasion is unclear, it is thought to occur in 8.3% to 40.0% of cases [5,6,7,8]. Estimating the risk of medullary invasion is critical for preoperative planning to achieve a negative margin. Because the tumour is so rare, few reports have examined its clinical, radiological and histological characteristics [2,5,8]. It is important to collect data and analyse characteristics and treatment outcomes if we are to improve the prognosis. Furthermore, only a few studies report histological evaluation of periosteal chondrosarcoma [3,6].

In this case study, we investigate the clinical and histological characteristics, treatments and clinical outcomes of seven patients with periosteal chondrosarcoma, focusing particularly on the risk of medullary invasion.

## 2. Materials and Methods

### Case Selection

Thirty-three patients diagnosed with a periosteal cartilaginous tumour were retrieved from the bone and soft tissue tumour archives (1998–2020) of the three participating institutions. The records were reviewed retrospectively and seven patients with pathologically proven periosteal chondrosarcoma were identified. The following clinical data were extracted from the records: (1) demographics and preoperative variables (age, sex, symptoms, anatomical site of the tumour (diaphysis, metaphysis, or epiphysis), tumour size and radiological features); (2) therapeutic variables (surgical margin, reconstruction); and (3) oncologic outcome (histological features, grade, medullary invasion, follow-up time, local recurrence and metastasis). Symptoms (presence or absence of pain or swelling) at the time of presentation were recorded, as was symptom duration (defined as the interval between onset of symptoms and the date of surgery). Radiological features were assessed using conventional radiographs, computed tomography (CT) and magnetic resonance imaging (MRI). Conventional radiographs and CT evaluation included an assessment of tumour shape, lesion mineralisation and cortical erosion or thickening. Tumour size was measured on preoperative MRI, and intrinsic signal characteristics, soft tissue oedema and medullary invasion were analysed. All cases were reviewed and confirmed by two experienced bone tumour surgeons and pathologists. Although histological grading of periosteal chondrosarcoma is not part of the 2020 World Health Organisation classification, it was performed in accordance with the methods used for conventional chondrosarcoma [1]. Follow-up was available for all patients, bar one (Case 1). Follow-up information included residual symptoms, local recurrence, metastasis and additional treatment of the lesion. In most cases, local recurrence was assessed using conventional radiographs, and metastasis was assessed by chest CT. 

## 3. Results

Pathological review revealed that seven of the 33 patients had been diagnosed with periosteal chondrosarcoma, whereas the remaining 26 were diagnosed as periosteal chondroma. Patient characteristics are summarised in Table 1. All were male, and the median age at diagnosis was 36 years (range, 22–49 years). At presentation, common symptoms included a painless (four patients) or painful (two patients) swelling. The average duration of pain and swelling was 3.0 months (range, 1–6 months) and 3.5 months (range, 1–12 months), respectively. No patients had limited range of motion in the joints.

All patients underwent conventional radiography and MRI; six tumours were also assessed by CT. The metaphysis of the distal femur was the most common tumour site (*n* = 3, 42.9%), followed by the metaphysis of the proximal humerus (*n* = 2, 28.6%) (Table 1). The mean tumour size (as measured by MRI) was 5.4 cm in the long axis (range, 2.0–8.8 cm); two tumours were smaller than 3.0 cm. Conventional radiography and CT revealed that all tumours arose from the surface of the bone; also, soft tissue masses with a calcified matrix were common (Figure 1a,b). Other findings included cortical erosion (*n* = 2), cortical thickening (*n* = 3) and a periosteal shell (*n* = 7). For all tumours, T1-weighted MRI showed homogeneous low signal intensity masses, whereas T2-weighted MRI showed predominantly high signal intensity masses. In three cases (42.9%), medullary invasion was observed on T2-weighted MRI (Figure 1c,d). Other characteristics included soft tissue oedema (*n* = 4).

Of the five cases that underwent open biopsy, three were diagnosed as periosteal chondrosarcoma and all were resected with a wide margin. Of the two cases that did not undergo open biopsy, one (Case 1) was resected with a wide margin because cortical erosion and soft tissue oedema were observed, and malignancy was clinically suspected. In the other case (Case 4), preoperative imaging made it difficult to distinguish benign from malignant. Therefore, the fibula and the extra-skeletal lesion were resected en bloc and with a marginal margin, respectively. To reconstruct the bone defect after tumour resection, two patients received endoprostheses, two received an autogenous bone graft and one received beta-tricalcium phosphate (β-TCP). Two patients did not require reconstruction of the bone defect because they had a tumour in the fibular diaphysis and pubis, respectively. 

Histologically, all tumours showed lobular proliferation of atypical chondrocytes with enlarged nuclei; double-nucleated cells and moderately increased cellularity were also observed. Three patients were diagnosed with periosteal chondrosarcoma grade I (42.9%) and the other four were diagnosed with grade II (57.1%). A cortical bone-permeating pattern was seen in all tumours; however, this was not observed in 26 periosteal chondromas (Figure 2). Furthermore, six tumours (85.7%) extended into the medullary cavity (Table 2). Of note, three (Cases 1, 4 and 6) of these did not show invasion of the medullary cavity on MRI evaluation (Figure 3), but invasion was confirmed upon pathological assessment (Figure 4); in all other cases (Cases 2, 3 and 5), invasion was confirmed by both MRI and pathological assessment (Figure 1c,d and Figure 2c,d). The mean distance from the cortex to the deepest lesion of the intramedullary tumour was 12.8 mm (range, 1.0–25 mm) (Table 2). Furthermore, of the six patients with medullary invasion, three had some intramedullary skip lesions (Cases 3, 4 and 5) (Figure 2d and Figure 4a,b); in the remaining three cases (Cases 1, 2 and 6), the intramedullary lesions were continuous with the periosteal tumours (Table 2). The average follow-up period for the seven patients was 58.6 months (range, 1–134 months). None of the seven patients had residual symptoms, local recurrence, metastasis, or additional treatment of the lesion.

## 4. Discussion

Here, we reviewed seven patients with periosteal chondrosarcoma and present the clinical and pathological characteristics, treatments and outcomes. Of note, careful pathological evaluation revealed that the rate of medullary invasion in this cohort (6/7 cases; 85.7%) was higher than that reported previously [5,6,7,8].

Generally, chondrosarcoma is resistant to chemotherapy and radiation therapy; therefore, surgery is the first treatment of choice. Resection with an adequate margin is strongly recommended for periosteal chondrosarcoma because reports suggest that inadequate excision results in local recurrence and pulmonary metastasis [2,3,9]. At surgical planning, the decision about the resection area is the most important issue to be addressed if a negative surgical margin is to be achieved. Although medullary invasion occurs occasionally, and can be one of the key factors in the decision regarding the surgical margin, few reports have examined this [5,6,7,8]. In addition, previous reports focused only on either imaging or histological evaluation of medullary invasion [5,6,8]. Therefore, this study used both MRI and histology to investigate medullary invasion. We found that the frequency of medullary invasion (85.7%) was higher than that reported previously (Table 3), suggesting that medullary invasion by periosteal chondrosarcoma is more common than previously thought. Of note, intramedullary tumours sometimes exhibited skip lesions from the main periosteal tumours. Furthermore, half of the histologically invasive tumours did not show invasion on MRI; therefore, surgeons should recognise that even if MRI does not show medullary invasion, it may actually be present on microscopic examination, and may extend more than 10 mm into the cortex.

The main differential diagnoses for periosteal chondrosarcomas are periosteal chondroma, periosteal osteosarcoma, or secondary peripheral chondrosarcoma. Periosteal chondroma is the most difficult tumour to differentiate from periosteal chondrosarcoma because the two have many overlapping features on imaging. Periosteal chondroma causes no cortical destruction, whereas periosteal chondrosarcoma often shows cortical destruction [2], although only two tumours (25%) in the present study showed cortical erosion on conventional radiographs or CT images. Furthermore, the size of the tumour is a reliable distinguishing parameter: periosteal chondrosarcomas are generally larger than 3 cm in diameter [5]. However, two out of seven tumours reported herein were smaller than 3 cm (Cases 1 and 3). In such cases, histological evaluation is important. Cellular atypia and a permeating pattern, which are observed only in periosteal chondrosarcoma, are promising histological findings. Periosteal osteosarcoma is an intermediate-grade malignant tumour arising on the surface of bone. On conventional radiographs, the tumour usually presents as a periosteal reaction perpendicular to the long axis of the bone, often with a sunburst or “hair-on-end pattern” [1,10]. Although periosteal osteosarcoma cells predominantly produce a chondroid matrix, identification of osteoid-producing areas with cartilaginous elements is useful for differentiating the tumour from periosteal chondrosarcoma [1]. Furthermore, heterozygous mutation of isocitrate dehydrogenase (IDH) is observed in about half of cartilage tumours, including periosteal chondrosarcoma, but not in osteosarcoma [11,12]. Therefore, detection of IDH mutations helps to distinguish periosteal chondrosarcoma from periosteal osteosarcoma, which produces a chondroid matrix. Also, it is often difficult to distinguish periosteal chondrosarcoma from secondary peripheral chondrosarcoma. In secondary peripheral chondrosarcoma, the medullar of the underlying bone is continuous with that of the stalk of the lesion, whereas periosteal chondrosarcoma originates from the periosteum [6]. Here, the different signal characteristics on MRI were useful for distinguishing these two tumours.

The study has several limitations. First, data from patients with periosteal chondrosarcoma were collected from three different hospitals. There were potential differences in diagnostic methods and treatment philosophies, although all the patients were treated by expert surgeons specialised in sarcoma. Also, all data were reviewed by the first and the corresponding authors. Second, the relatively small number of patients limits the power of the study. We performed a multi-institutional study because periosteal chondrosarcoma is extremely rare. We believe that this study provides useful information about tumour characteristics as it is one of very few to examine this rare tumour in detail.

In summary, we present a rare cases series comprising seven patients with periosteal chondrosarcoma in which we evaluated clinical and pathological characteristics, treatments and outcomes. The rate of medullary cavity invasion was higher than that reported previously. Therefore, a wide resection is recommended, and surgeons should pay close attention to the possibility of medullary invasion, even if the tumour does not invade the medullary cavity on MRI.

## 5. Conclusions

We evaluated medullary invasion of periosteal chondrosarcoma both radiologically and histologically and found that the frequency was higher than that reported previously. Therefore, surgeons should consider the possibility of medullary invasion at surgery planning in order to achieve a histologically negative margin, even if the tumour does not show invasion into the medullary cavity on MRI.

## Figures and Tables

**Figure 1 jcm-11-02062-f001:**
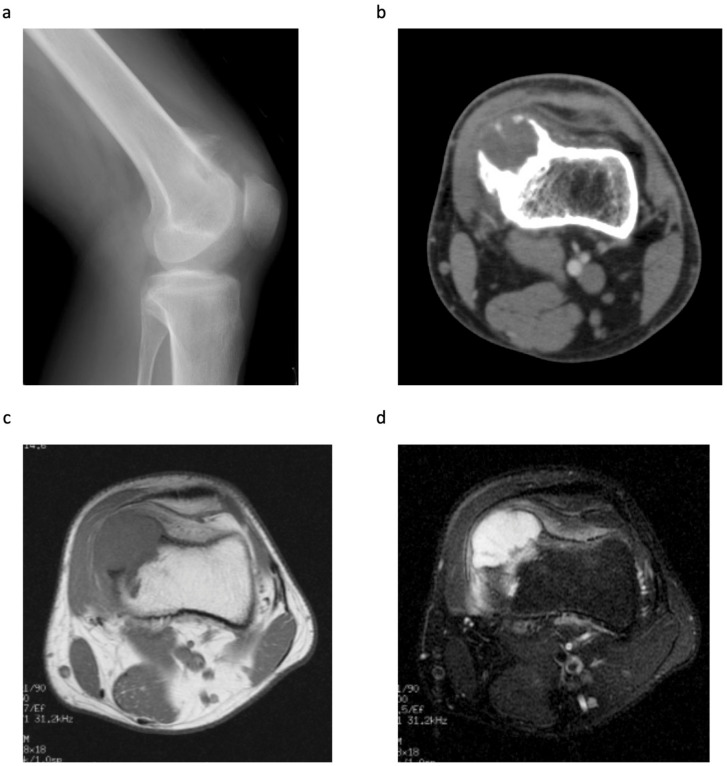
A 23-year-old male with a pathological diagnosis of periosteal chondrosarcoma (Case 3). (**a**) Lateral radiograph of the left knee showing a lobulated calcified mass on the anterior aspect of the distal metaphysis of the femur. (**b**) CT showing the medial periosteal-based lesion with a calcified shell. The cortex is thickened, but there is no evidence of medullary invasion. (**c**) T1-weighted MRI showing a lobulated mass (6.0 × 4.0 × 3.0 cm), with low signal intensity, arising from the periosteum. (**d**) T2-weighted fat suppressed MRI showing a predominantly high signal intensity mass with medullary invasion. Adjacent soft tissue oedema is also present.

**Figure 2 jcm-11-02062-f002:**
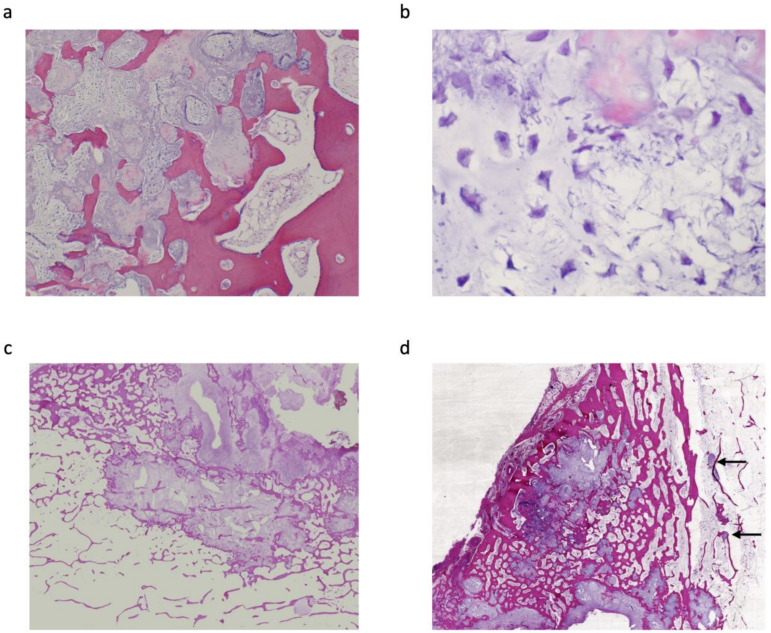
Histopathological sections (haematoxylin & eosin stain) from Case 3; a grade II periosteal chondrosarcoma. (**a**) Section showing proliferation of atypical chondrocytes, embedded in a cartilaginous matrix, accompanied by myxoid changes. Permeation of the cortical bone is also prominent (×40). (**b**) Tumour cells with hyperchromatic nuclei embedded in the myxoid matrix (×300). (**c**) Atypical chondrocytes invading the medullary cavity (×20). (**d**) Skip lesions from the periosteal tumour invading the medullary cavity (black arrow).

**Figure 3 jcm-11-02062-f003:**
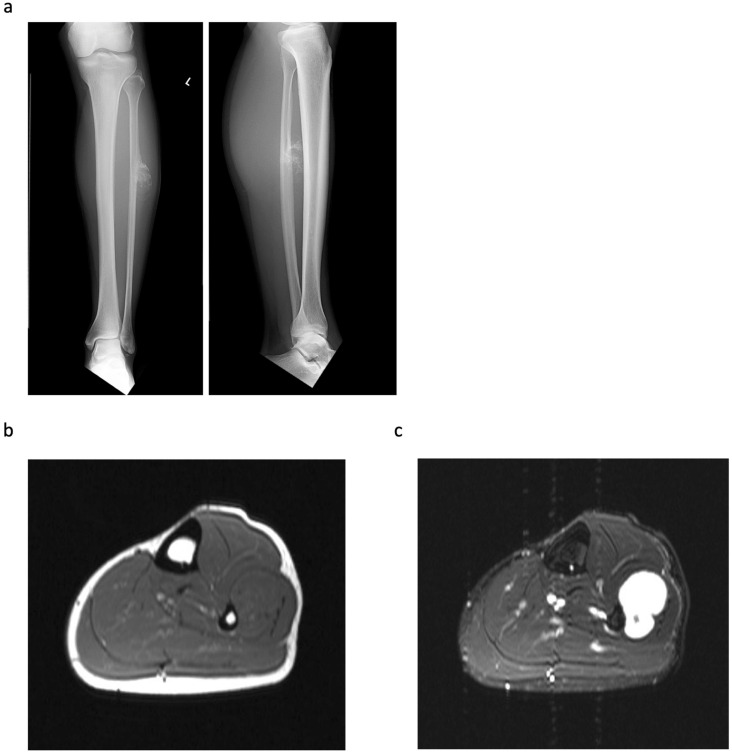
A 37-year-old male with a pathological diagnosis of periosteal chondrosarcoma (Case 4). (**a**) Radiograph of the left leg showing a lobulated calcified mass on the anterior aspect of the diaphysis of the fibula. The cortex is eroded. (**b**) T1-weighted MRI showing a lobulated mass (5.5 × 2.5 × 3.5 cm), with low signal intensity, arising from the periosteum. (**c**) T2-weighted fat suppressed MRI showing a predominantly high signal intensity mass. Neither medullary invasion nor adjacent soft tissue oedema are present.

**Figure 4 jcm-11-02062-f004:**
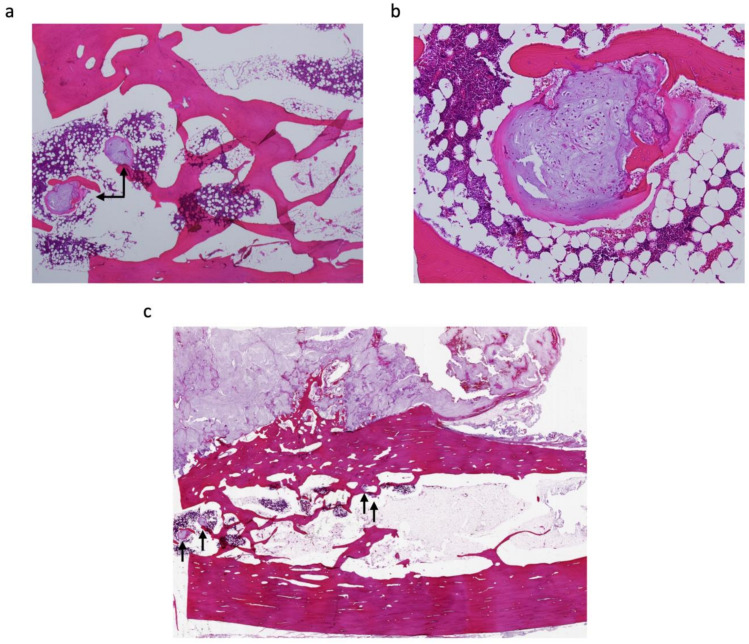
Histopathological sections (haematoxylin & eosin stain) from Case 4; a grade I periosteal chondrosarcoma. (**a**) Tumour cells invading the medullary cavity (black arrow) (×20). (**b**) Atypical chondrocytes embedded in the myxoid matrix (×100). (**c**) Skip lesions from the periosteal tumour invading the medullary cavity (black arrow).

**Table 1 jcm-11-02062-t001:** Clinical characteristics of the seven patients with periosteal chondrosarcoma.

Case	Sex	Age	Symptoms	Duration of Symptom (Months)	Site	Size (cm)	Cortex	Surgical Margin	Reconstruction	Histological Grade	Follow-Up Period (Months)	Recurrence	Metastasis
1	M	22	pain swelling	1 1	distal femur metaphysis	2 × 1 × 2.5	erosion	wide	β-TCP	I	1	−	−
2	M	36	swelling	2	distal femur metaphysis	2 × 1.5 × 2	no change	wide	bone autograft + plate	II	72	−	−
3	M	23	pain swelling	6 3	distal femur metaphysis	6.0 × 4.0 × 3.0	thickening	wide	DFR	II	134	−	−
4	M	37	swelling	12	fibula diaphysis	5.5 × 2.5 × 3.5	erosion	marginal	−	I	59	−	−
5	M	41	pain	2	proximal humerus metaphysis	5.2 × 4.8 × 4.4	no change	wide	PHR	II	77	−	−
6	M	49	swelling	2	pubis	8.8 × 5.4 × 5.4	thickening	wide	−	I	36	−	−
7	M	35	swelling	1	proximal humerus metaphysis	7.5 × 5.5 × 5.5	thickening	wide	bone autograft + plate	II	32	−	−

β-TCP: beta-tricalcium phosphate; DFR: distal femur replacement; PHR: proximal humeral replacement.

**Table 2 jcm-11-02062-t002:** Details of medullary invasion by periosteal chondrosarcoma.

Case	Medullary Invasion (MRI)	Medullary Invasion (Pathological)	Distance from the Cortex to the Deepest Lesion of the Intramedullary Tumour (mm)	Skip or Continuous from Periosteal Tumour
1	−	+	15	continuous
2	+	+	15	continuous
3	+	+	25	skip
4	−	+	3	skip
5	+	+	18	skip
6	−	+	1	continuous
7	−	−		

**Table 3 jcm-11-02062-t003:** Comparison of previous studies of periosteal chondrosarcoma with invasion of the medullary cavity.

Reference	Rate of Medullary Invasion
Robinson et al. [5]	9.1% (2/22)
Cleven et al. [6]	40.0% (4/10)
Hatano et al. [7]	Case report
Vanel et al. [8]	8.3% (2/24)
Current study	85.7% (6/7)

## Data Availability

All data generated or analysed during this study is included in this published article.

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
