# Peer review of "Clinical, Radiological, and Histopathological Characteristics of Periosteal Chondrosarcoma with a Focus on the Frequency of Medullary Invasion"

_jcm, 2022, doi:10.3390/jcm11072062_

Round 1
Reviewer 1 Report
This paper is interesting because it describes a very rare disease called periosteal chondrosarcoma. It is educational because it describes a case in which invasion was not suspected on MRI but was confirmed on pathology.
Two patients did not undergo open biopsy, one with extensive resection and the other with marginal resection. Was this because one case was clinically suspected to be malignant and the other was judged to be benign? Please explain in more detail how this is caused by the difference in treatment concepts.
Author Response
Dear Editor and Reviewer #1:
We are grateful to reviewer #1 for their critical comments and useful suggestions, which helped us improve the paper. As indicated in our responses that follow, we have taken all of these comments and suggestions into account in the revised version of the manuscript.
Comment #1
Two patients did not undergo open biopsy, one with extensive resection and the other with marginal resection. Was this because one case was clinically suspected to be malignant and the other was judged to be benign? Please explain in more detail how this is caused by the difference in treatment concepts. 
Response
We thank the reviewer for bringing this important point to our attention. As we described the limitation in discussion (Line 225 – 232 in the original paper), we analyzed a retrospective data collected from three different hospitals and there were potential differences in diagnostic methods and treatment strategy.
The cases in which open biopsy was not performed in our study were case 1 and case 3.
In case 1, the patient was seen as early as one month after becoming aware of the pain at distal femur. In addition, cortical erosion and soft tissue oedema were observed on CT and MRI, respectively. Therefore, although the tumor was less than 3 cm in diameter at the time of visit, the possibility of periosteal chondrosarcoma was clinically suspected, and an extensive resection was performed.
On the other hand, case 4 was an unusual tumor arising from fibula. CT showed cortical erosion, but MRI did not show any signal change in the soft tissue around the tumor. Therefore, preoperative imaging findings made it difficult to distinguish benign tumor from malignant one. Then, the fibula and the extra-skeletal lesion were resected en-bloc and with a marginal margin, respectively (Line 128 – 130 in the original paper). Although medullary invasion was observed histologically in case 4, we had an adequate margin from the tumor in the fibula, and neither recurrence nor metastasis were observed.
In the revised manuscript, we have added the following text on Lines 128–129: “because cortical erosion and soft tissue oedema were observed, and malignancy was clinically suspected”, and on Lines 130–131: “preoperative imaging made it difficult to distinguish benign from malignant. Therefore,” , respectively.
We look forward to hearing from you regarding the submitted manuscript. We would be glad to respond to any further questions and comments that you might have.
Sincerely,

Reviewer 2 Report
Congratulation for the well conducted multicentric work, and for the clear presentation of your results.
I think it's relevant to describe medullary invasion and skip lesions (rare events) in periosteal chondrosarcoma, (a very rare disease).
The strenghts and weaknesses of the work are also exhaustively discussed
Author Response
We would like to thank you for your time and efforts in reviewing our manuscript. Your positive comments are appreciated.
Sincerely,

Reviewer 3 Report
Nakagawa and collagues describe a series of 7 cases of periosteal chondrosarcoma focusing on clinical, radiological, and histopathological features. The authors underline the importance of medullary invasion when attempting to achieve a histologically negative margin, even if the tumour does not show invasion into the medullary cavity on MRI. This data has just been known in other bone superficial tumor (see Giapponesi e Vanel) and the high percentage of medullary invasion is surprising comparing to other series reported in literature (see table 3).
The number of cases (7) is too low to support the conclusions. Moreover could be interested to evaluate IDH1 and IDH2 mutations and to compare to periostel chondroma cases.
Other comments: How do the authors distinguish periosteal chondroma to periosteal chondrosaroma? only with the tumor size?
Figure 4 is not so convincing as medullary invasion. I would like to see the tumoral mass with medullary invasion to low power magnfication.
Round 2
Reviewer 3 Report
the authors replied well to my comments/suggestions. The low number of cases remain the major criticism.